# Carbon and Manganese in Semi-Insulating Bulk GaN Crystals

**DOI:** 10.3390/ma15072379

**Published:** 2022-03-23

**Authors:** Mikolaj Amilusik, Marcin Zajac, Tomasz Sochacki, Boleslaw Lucznik, Michal Fijalkowski, Malgorzata Iwinska, Damian Wlodarczyk, Ajeesh Kumar Somakumar, Andrzej Suchocki, Michal Bockowski

**Affiliations:** 1Institute of High Pressure Physics, Polish Academy of Sciences, Sokolowska 29/37, 01-142 Warsaw, Poland; zajac@unipress.waw.pl (M.Z.); tsochacki@unipress.waw.pl (T.S.); bolo@unipress.waw.pl (B.L.); felix@unipress.waw.pl (M.F.); miwinska@unipress.waw.pl (M.I.); bocian@unipress.waw.pl (M.B.); 2Institute of Physics, Polish Academy of Sciences, Aleja Lotników 32/46, 02-668 Warsaw, Poland; wlodar@ifpan.edu.pl (D.W.); skumar@ifpan.edu.pl (A.K.S.); suchy@ifpan.edu.pl (A.S.); 3Center for Integrated Research of Future Electronics, Institute of Materials and Systems for Sustainability, Nagoya University, C3-1 Furo-cho, Chikusa-ku, Nagoya 464-8603, Japan

**Keywords:** gallium nitride, halide vapor phase epitaxy, co-doping, carbon, manganese

## Abstract

Co-doping with manganese and carbon was performed in gallium nitride grown by halide vapor phase epitaxy method. Native seeds of high structural quality were used. The crystallized material was examined in terms of its structural, optical, and electrical properties. For that purpose, different characterization methods: x-ray diffraction, Raman spectroscopy, low-temperature photoluminescence, and temperature-dependent Hall effect measurements, were applied. The physical properties of the co-doped samples were compared with the properties of crystals grown in the same reactor, on similar seeds, but doped only with manganese or carbon. A comparison of the electrical and optical properties allowed to determine the role of manganese and carbon in doped and co-doped gallium nitride crystals.

## 1. Introduction

One of the most important applications of gallium nitride (GaN) and aluminum nitride (AlN), as well as their ternary alloys, Al_x_Ga_1−x_N, is a high electron mobility transistor (HEMT) [1,2]. Its epitaxial structure (GaN/AlGaN) is mainly crystallized on a foreign semi-insulating (SI) substrate as silicon carbide (SiC) or silicon (Si) [3]. Such a structure demonstrates a high threading dislocation density (TDD) due to a relaxation process of the nitride layers on a foreign wafer of different lattice parameters [4]. It is commonly believed that in the case of the HEMT epitaxial structures, the high TDD (of the order of 10^8^ cm^−2^) is of no importance and does not interfere with the operation of the device. There are reports that better electrical parameters (higher current densities, lower three-terminal OFF-state leakage, and lower current collapse) can be obtained by building HEMTs on native GaN substrates [5]. However, they need to be available in large sizes (2 or 4 inches in diameter) and, at the same time, with a low price. Otherwise, HEMT structures will be grown on foreign substrates. Semi-insulating GaN wafers are fabricated from crystals grown by two methods: ammonothermal and halide vapor phase epitaxy (HVPE) [6,7,8,9,10,11,12]. Applying SI ammonothermal GaN (Am-GaN) in the industry is rather impossible due to the lack of mass production of 2-inch SI wafers. In turn, 2- and 4-inch SI HVPE-GaN substrates are commercially available. They are obtained by wafering from HVPE-GaN crystals doped with iron (Fe). Semi-insulating Am-GaN is obtained by doping GaN with manganese (Mn) in order to compensate the oxygen donors [13].

The resistivity of HVPE-GaN:Fe wafers and crystals is of the order of 10^7^ Ωcm at room temperature (RT) [14,15]. A much higher resistivity was presented for HVPE-GaN doped with Mn or carbon (C) [15,16]. The value exceeded 10^8^ Ωcm even at 300 °C. In order for a HEMT device to function properly, the substrate’s resistivity must be of the order of 10^8^ Ωcm at 300 °C. Therefore, HVPE-GaN:Mn or HVPE-GaN:C becomes the obvious choice for this application. Carbon seems to be an easy dopant in terms of selecting an appropriate precursor and introducing it into the HVPE reactor. The use of methane (CH_4_), ethane (C_2_H_6_), or acetylene (C_2_H_2_) as the C precursor is not a technological challenge [17]. It was shown that HVPE-GaN:C crystals are of high structural quality and can be grown at a high rate, of the order of 200 µm/h [18,19]. Mechanical properties of HVPE-GaN:C are similar to those of unintentionally doped (UID) HVPE-GaN. Therefore, all the wafering procedures needed for preparing substrates from crystals can be applied in a relatively easy way. Carbon substituting nitrogen (C_N_) acts as a deep acceptor with a transition energy level at around 1 eV above the valence band maximum (vbm) [20]. It was, however, shown that C is self-compensated in HVPE-GaN [21,22,23]. The compensation ratio of C depends on its concentration in GaN samples and for high values (>10^19^ cm^−3^) the compensation can be equal to 1 [23]. Different explanations of the compensation were provided. One of them is based on a theoretical prediction for C on a Ga site acting as a shallow donor [20].

As mentioned, HVPE-GaN:Mn crystals also demonstrate high resistivity at RT or 300 °C [15]. However, there is no good gas precursor for Mn in the HVPE technology. Thus, solid Mn is usually placed in the low-temperature zone of the HVPE reactor. Hydrochloride (HCl) has to flow above the metal and react with it in order to form manganese chloride (MnCl_x_) which is transported to the growth zone. The structural quality of HVPE-GaN:Mn is high and the growth rate can reach even 200 µm/h [15]. However, HVPE-GaN:Mn, similar to Am-GaN:Mn, is very fragile and applying the wafering procedures for obtaining substrates from crystals with a proper high yield is challenging.

It would be hard to obtain GaN of higher resistivity than GaN:Mn. The ${\mathrm{Mn}}^{2+/3+}$ energy transition level lies close to the middle of the GaN band gap [24,25]. On the other hand, according to theoretical predictions [26], Mn may also exist in the Mn^3+/4+^ level. This state can, theoretically, be obtained by co-doping of Mn with other acceptors. However, the existence of Mn^3+/4+^ in bulk GaN crystals has not been confirmed experimentally so far. In this work, Mn^3+/4+^ will be detected in GaN doped with Mn and C. Therefore, a co-doping process of HVPE-GaN by Mn and C was performed. Moreover, co-doping with C could change some mechanical properties of SI HVPE-GaN and facilitate the wafering procedures. Before checking the mechanical properties and launching the fabrication of new co-doped GaN wafers, physical (structural, optical, and electrical) properties of HVPE-GaN:Mn,C should be examined and analyzed. The role of C in SI HVPE-GaN:Mn,C was investigated. The behavior of Mn in GaN co-doped with C was also analyzed. Eight crystal growth experiments were carried out. Two of them were devoted to HVPE-GaN:C crystallization to serve as a reference. These experiments were performed in the same way as previously published [17]. At constant technological parameters used in a typical UID HVPE-GaN growth, two flows of CH_4_ were applied. Three reference growth runs were also dedicated to HVPE-GaN:Mn [15]. Again, all the parameters were kept constant with values typical for UID HVPE-GaN growth and three different flows of HCl over Mn were applied. Next, three processes were performed to obtain HVPE-GaN:Mn,C crystals. For a constant HCl stream over solid Mn, the flow of CH_4_ was changed. The morphology, structural quality, growth rate, as well as electrical and optical properties of the new-grown crystals were examined, analyzed, and then discussed.

## 2. Materials and Methods

The HVPE method is the crystallization from the gas phase. Hydrochloride reacts with liquid gallium at relatively low temperatures (800–900 °C) forming gallium chloride (GaCl). In this temperature range, the partial pressure of GaCl is much higher than that of GaCl_3_. Therefore, the formation of GaCl is more favorable [27]. The latter is transported by the carrier gas (CG, mainly hydrogen or nitrogen) to the crystal growth zone (at a temperature of 1000–1100 °C) where a seed is placed. Herein, GaCl reacts with ammonia (NH_3_) to synthesize GaN. A silica glass, horizontal home-made HVPE reactor, described in detail elsewhere [15], was applied for crystallizing GaN:C, GaN:Mn, and GaN:Mn,C. The C precursor, CH_4_, was introduced through an additional pipe placed in the gallium chloride (GaCl) nozzle as shown in Figure 1. Solid Mn was placed in a quartz boat in the low-temperature zone of the reactor. Hydrochloride (1% diluted in hydrogen) was flown over Mn and MnCl_x_ was transported in a line parallel to the GaCl one (see Figure 1). These two lines were connected before reaching the seed, forming a Y-shaped tube. The reactants’ flows of the eight crystallization runs are presented in Table 1. Hydrogen (H_2_) was used as the carrier gas. The V/III ratio for all processes was 20. The crystal growth temperature was always 1320 K and the crystallization time was 3.5 h.

One-inch n-type Am-GaN epi-ready wafers, misoriented by 0.3° in the $\left[10\bar{1}0\right]$ direction, were used as seeds. After the growth series, the morphology of the as-grown HVPE-GaN was studied by optical microscopy with differential interference contrast (DIC). Then, $\left(002\right)$ symmetrical reflection X-ray rocking curves (omega scan) were measured using a Philips high-resolution X’Pert Pro diffractometer equipped with a four-reflection Bartels monochromator (Philips Analytical, Almelo, The Netherlands). The size of the X-ray beam was 1 mm × 10 mm (width in the diffraction plane was 1 mm). For further characterization, the seeds were removed by mechanical polishing, leaving freestanding HVPE-GaN crystals that were lapped and polished to an optically flat state on both sides. The crystals were then diced into a few smaller and 300-µm-thick square samples (5 mm × 5 mm) for different characterization methods applied on the $\left(0001\right)$ and $\left(000\bar{1}\right)$ planes. Secondary ion mass spectrometry (SIMS) was used to investigate the concentrations of impurities on both sides of the prepared samples.

Raman spectroscopy was applied on the $\left(0001\right)$ planes for all the co-doped samples. Moreover, a reference UID GaN sample was also measured [28,29]. Confocal micro-Raman measurements were carried out using a Monovista CRS+ spectrometer equipped with a single-mode Cobolt Samba 532 nm laser, a 0.75 m Acton–Princeton monochromator, 2400 grooves/mm holographic grating, and a Princeton Instruments back-thinned, deep-depleted, nitrogen-cooled CCD (1340 × 100 pixel array) camera (Photonics Media/Laurin Publishing Co., Inc., Pittsfield, MA, USA). The spectral resolution of the system was around 0.5 cm^−1^, while the minimum step was around 0.25 cm^−1^. The laser power was attenuated using neutral density filters to perform measurements at approximately 4.5 mW/µm^2^. A notch filter was placed to filter out the laser light with a cutting edge at approximately 60 cm^−1^ away from the laser line. The samples were mounted on a precise, computer-controlled Olympus XYZ, IX71 microscope table. The laser was focused using a 50× Olympus objective with a numerical aperture of 0.50. All the measurements were performed in a backscattering, $z\left(xx\right)\bar{z}$, configuration at RT.

Low-temperature photoluminescence (LTPL) spectra were obtained at 8 K using a 325 nm He–Cd laser (Kimmon Koha Co., Ltd., Fukushima, Japan) with a power density of about 5 W/cm^2^ as the excitation source. The diameter of the spot was around 200 µm. Emission from the sample, collected in a backscattering geometry and dispersed by a SPEX500M spectrometer (SPEX LAB, Metuchen, NJ, USA), was detected by a photomultiplier (Hamamatsu Photonics tube r943-02).

Resistivity ρ and Hall constant R_H_ were measured in the van der Pauw configuration at temperatures reaching 1000 K. The samples were prepared by placing Ni(250 Å)/Au(750 Å) contacts on the (0001) surface and annealed at 773 K for a few minutes under an atmosphere of N_2_ with a 20% admixture of O_2_. The quality of the contacts enabled measurements of ohmic resistivity in the studied temperature range. The measurement parameters, such as operating current, were controlled to ensure ohmic conditions.

## 3. Results

The morphology of all the grown crystals was identical. One or a few hillocks were observed on the as-grown surfaces. Such kind of morphology has already been described elsewhere [28,30]. The X-ray measurements showed that almost all the new-grown GaN crystals were of the highest structural quality, similar to that of the applied Am-GaN seeds. The values of the full width at half maximum (FWHM) for the $\left(002\right)$ reflection were between 20 arcsec and 40 arcsec and the bowing radii (R) of the crystallographic planes were above 10 m for samples #1 and #2. The structural quality of the crystals doped with Mn was slightly worse—the FWHM value was between 50 arcsec and 100 arcsec and R was around 8 m. It was observed that, in the case of the co-doped samples #6 and #7, FWHM was at the level of 40–60 arcsec and R was above 10 m. Crystal #8 had the worst structural quality (based on the X-ray diffraction (XRD) results)—its FWHM was above 500 arcsec and R was below 1 m. In general, for crystals doped with C or Mn, the growth rate did not differ from the typical growth of UID HVPE-GaN grown on Am-GaN substrates in the same conditions, taking the value from 170 µm/h to 200 µm/h [30]. The growth rate of the co-doped crystals was around 130 µm/h.

Table 2 presents the SIMS data obtained for the $\left(0001\right)$ plane of the prepared samples. It is worth mentioning that no difference was observed in the concentrations of the examined impurities on both the $\left(0001\right)$ and $\left(000\bar{1}\right)$ sides. The concentrations of the main donors (silicon (Si), oxygen (O), as well as hydrogen (H)) were 1 × 10^17^ cm^−3^ or lower (the background level of (Si), (O), and (H) is 5 × 10^16^ cm^−3^, 1 × 10^16^ cm^−3^, and 1 × 10^16^ cm^−3^, respectively). For HVPE-GaN:C, no traces of Mn were detected (the value was below the background level—5 × 10^15^ cm^−3^). The carbon impurity concentration (C) varied from 1 × 10^19^ cm^−3^ to 5 × 10^19^ cm^−3^ depending on the CH_4_ flow. For HVPE-GaN:Mn, the (C) was lower than the SIMS background level (2 × 10^16^ cm^−3^). The concentration of the Mn impurity varied from 2 × 10^18^ cm^−3^ to 5 × 10^18^ cm^−3^ depending on the HCl flow over solid Mn. In the co-doped samples, the (Mn) was constant and the (C) rose with increasing the flow of the precursor (CH_4_).

Figure 2a presents a typical Raman spectrum taken from the $\left(0001\right)$ surface in the backscattering geometry for a reference UID HVPE-GaN sample. Two peaks always detected for such material were: E_2_^high^ at 567.5 cm^−1^ and A_1_(LO) at 733.5 cm^−1^. These modes were also well visible in the other measured GaN samples. For all the obtained crystals, we did not observe any changes in the position of the E_2_^high^ and A_1_(LO) modes. During the examination of the doped material, we focused on the differences between the obtained spectra. Thus, Figure 2b–d show the Raman spectra for HVPE-GaN:C, HVPE-GaN:Mn, and HVPE-GaN:Mn,C in the range between 600 cm^−1^ and 900 cm^−1^. The presented area covered the vicinity of the A_1_(LO) mode. This was the region where the major differences between the examined samples could be seen. The Raman spectra of samples #1 and #2, presented in Figure 2b, revealed additional peaks ω_1_ and ω_2_ in comparison with UID HVPE-GaN. The positions of these phonon modes were insensitive to the amount of C in the examined samples and were equal to 766 cm^−1^ (ω_1_) and 775 cm^−1^ (ω_2_). Additional modes, marked in Figure 2b as A, B, and C at 640 cm^−1^, 659.5 cm^−1^, and 672 cm^−1^, respectively, appeared in the Raman spectra.

The Raman spectra of crystals doped with Mn looked entirely different. One additional peak (marked in Figure 2c as D) was detected in position 667 cm^−1^. The intensity of this peak depended on the Mn concentration and increased with the Mn content. Additional features were detected in the Raman spectra of crystals #6, #7, and #8 co-doped with Mn and C (see Figure 2d). Some new peaks were found (marked as E and F). Mode D was also present in this spectrum. It should be noted that the position of mode D depended on the $\left[C\right]/\left[\mathrm{Mn}\right]$ ratio (see Table 3). In the co-doped GaN:Mn,C samples, another broad peak, E (at 692 cm^−1^), was found. The last peak was mode F at 770 cm^−1^ (on the right slope of mode A_1_(LO)). It can be seen that for sample #6, this peak was very sharp and intense. In the case of samples #3, #4, and #5, doped only with Mn, this mode was not visible. On the other hand, mode F was broad and jagged for sample #8 with a high amount of C. It is also worth noting that the position of this band was exactly between the positions of the ω_1_ and ω_2_ modes.

Figure 3 represents the results of LTPL for all samples. The spectra for samples #1 and #2 are typical for GaN:C crystals [18,22]. One can see a strong and broad yellow luminescence (YL) peak at around 2.4 eV. It can be seen that the shapes of the spectra are similar, but the intensities are rather different. For sample #2, with a higher C content, YL is more intense. Moreover, in the case of sample #2, YL is slightly shifted towards higher energies. The second component of the spectrum, marked in the graph, is aquamarine luminescence (AL) with a maximum at 2.59 eV. The presented spectra contain also one more component at around 3.0 eV: blue luminescence (BL). Additionally, no band edge (BE) LTPL peaks were detected for samples doped with C.

Figure 3b presents spectra for crystals doped with Mn. It can be seen that the three spectra differ from each other—the shape measured for sample #3 varies significantly from the other two samples (which spectra are essentially similar to each other, but with different intensities). The YL peak for sample #3 is shifted towards lower energy: 2.34 eV in comparison with 2.41 eV for samples #4 and #5. On the other hand, a broad BL line is in the same position for all samples: at 2.94 eV. In all the spectra measured for GaN:Mn, one can see BE peaks with low intensities. Line 3.48 eV can be interpreted as an ABE (acceptor bound exciton) emission. This peak can be correlated with a clearly visible donor–acceptor pair (DAP) emission line at 3.27 eV and LO phonon replicas (3.17 eV and 3.07 eV).

Figure 3c,d present PL spectra of crystals co-doped with Mn and C. All three spectra are similar, one can see very intense YL (at around 2.4 eV). The other intense luminescence is BL and it consists (as for samples doped only with Mn) of two components: 2.81 eV and 2.95 eV. In the energy range above 3.0 eV, one can observe many lines (Figure 3d). Similar to the case of the GaN:Mn samples, the origins of these lines are: ABE (at 3.46 eV), DAP (at 3.26 eV), and DAP-1LO phonon replica. In addition, a number of the so-called Y-lines (in the photon energy range between 3.1 and 3.5 eV) can be seen [31,32]. These luminescence lines were found in all co-doped samples. One can also find a peak from free excitons (at 3.51 eV). Additionally, on each of the spectra, one can find two sharp lines at 3.44 and 3.53 eV and their origin is the He–Cd laser used.

The results of the carrier concentration and resistivity as a function of inverse temperature for the co-doped samples are presented in Figure 4a,b, respectively. Due to very small carrier mobility, the Hall measurements were possible only for sample #8 at a very high temperature, close to 1000/T = 1 K^−1^ (Figure 4a).

In the fitting procedure, it was assumed that C plays the role of both donor and acceptor. The hole concentration and resistivity were modeled by a numerical solution of the charge neutrality equation in the form:(1)$$p+\;N_{D0}^{+}+N_{CGa}^{+}=n+N_{Mn}^{-}+N_{CN}^{-},$$
where: *p*, *n* are hole and electron concentrations, respectively (*n* in the case of *p*-type conductivity is neglected), $N_{D0}^{+}$ is the concentration of ionized residual (oxygen and silicon) donors, $N_{CGa}^{+}$ is the concentration of ionized $C_{Ga}$ donors, $N_{Mn}^{-}$ is the concentration of ionized Mn acceptors, and $N_{CN}^{-}$ is the concentration of ionized C acceptors. In the case of acceptors:(2)$$N_{i}^{-}=\;\frac{N_{i}}{1+g_{i}\cdot \exp\left(\frac{E_{i}-E_{F}}{k_{B}T}\right)},$$
where *i* stands for Mn or *C_N_*, *g_i_* is the degeneracy of energy level *i* and *E_F_* is the Fermi energy, *k_B_*—is the Boltzmann constant, and *T* is temperature. In the case of donors:(3)$$N_{D0}^{+}=N_{D0};\;N_{CGa}^{+}=N_{CGa},$$
where $N_{D0}^{+}$ and $N_{D0}$ are the concentrations of ionized and neutral residual (oxygen and silicon) donors, respectively, and $N_{CGa}^{+}$, $N_{CGa}$ are the concentrations of ionized and neutral *C_Ga_* donors, respectively. The fitting parameters are collected in Table 4. The input dopant concentrations were fixed and equal to the SIMS data from Table 2. Two cases were considered for fitting the concentration dependence from inverse temperature (see Figure 4a). The first one shows the case with the self-compensation ratio of C equal to 1. In the second fitting, it was assumed that all C acts as C_N_ acceptors. The resistivity was calculated using the same parameters and assuming the hole mobility µ_h_ (Table 4), which is independent of the temperature.

**Figure 4 materials-15-02379-f004:**
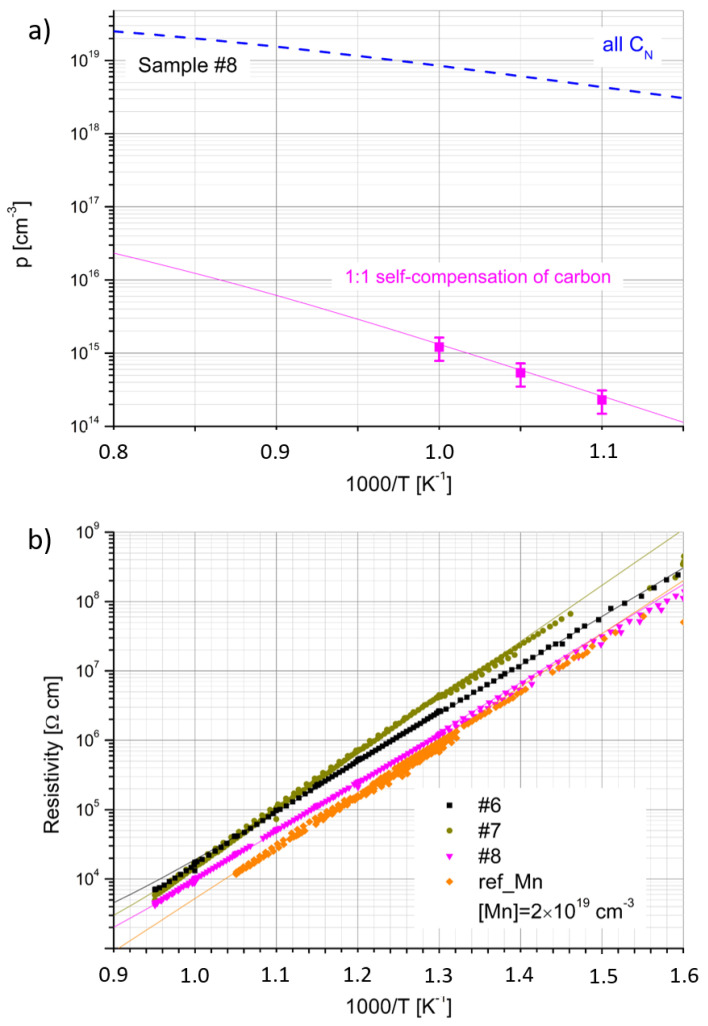
(**a**) Hole concentration (p) as a function of inverse temperature for GaN:Mn,C (sample #8); pink solid line represents the solution of charge neutrality equation using the self-compensation ratio of C equal to 1 and fitting parameters presented in Table 4; blue dashed curve represents similar modeling results, assuming all C acting as C_N_ acceptors; (**b**) resistivity of all GaN:Mn,C crystals; and solid lines represent the calculation of resistivity using modeled hole concentration (in panel (**a**)) and assuming mobility listed in Table 4. For comparison, a highly Mn-doped sample ref_Mn, taken from previous work [15], is presented.

The value of activation energy (*E_A_*), determined from the slopes of p(1/T) and ρ(1/T), was in the range between 1.4 eV and 1.6 eV. However, it turned out that for a very high Mn concentration (x_Mn_ = 2 × 10^19^ cm^−3^) [15], the *E_A_* determined from ρ(1/T) was also 1.4 eV. In this case, it was not possible to perform the Hall effect measurements. The ρ(1/T) dependence, measured for the highly doped reference Mn sample, taken from previous work, is also presented in Figure 4b. It can be seen that its resistivity is comparable (slightly lower) with that of sample #8 of clearly p-type conductivity. The results of resistivity measurements for HVPE-GaN:C were identical with the previously reported data (see [16,18]). The GaN:C samples were highly resistive and demonstrated p-type conductivity of temperature dependence characterized by *E_A_* of 1 eV.

**Table 4 materials-15-02379-t004:** The parameters used for the solution of the charge neutrality equation (Equations (1)–(3)) in order to fit experimental data presented in Figure 4. For sample ref_Mn (GaN:Mn), $N_{CGa}+N_{D0}\;=\;N_{D0}$ —the amount of C was below SIMS detection limit and therefore *N_CGa_* << *N_D0_*.

No.	*N_Mn_* (cm^−3^)	*N_CN_* (cm^−3^)	*N_CGa_*+*N_D0_* (cm^−3^)	*E_Mn_* (eV)	*E_CN_* (eV)	µ_h_ (cm^2^/Vs)
#6	3 × 10^18^	2 × 10^18^	2 × 10^18^	1.53	0.9	0.5
#7	2.5 × 10^18^	1 × 10^19^	1 × 10^19^	1.53	0.9	0.12
#8	3 × 10^18^	2 × 10^19^	2 × 10^19^	1.53	0.9	0.5
Ref_Mn	2 × 10^19^	-	3 × 10^17^ (N_D0_)	1.5	-	0.4

## 4. Discussion

It was demonstrated that it is possible to crystallize HVPE-GaN:Mn,C of very high structural quality on native ammonothermal seeds. Deterioration of the quality was observed only for crystal #8 with the highest concentration of C. The reason for this is not clear and requires further research. The next important result was that no gradient of the introduced dopants was observed in the grown HVPE-GaN. This means that uniform—in terms of dopant and impurities concentrations—crystals as well as wafers can be obtained. It is a very perspective result from the technological point of view. Homogenous crystal doping allows to fabrication of SI substrates with uniform electrical properties.

Raman scattering measurements showed large differences between the spectra obtained for the investigated doped and co-doped samples. Based on these results, one can predict the electrical states of the dopant in GaN or what complexes it forms. Crystals doped only with C revealed additional, in comparison with UID GaN, ω_1_ and ω_2_ modes. These two peaks result directly from C substituting nitrogen (C_N_) and are responsible for vibrations parallel and perpendicular to the c axis, respectively [33]. As was previously mentioned, positions of ω_1_ and ω_2_ were insensitive to the amount of C in the examined samples. The same situation was observed for B and C modes. The intensities of these additional peaks enhance with increasing the dopant concentration (see Table 2 and Figure 2b). These modes may be the result of the amphoteric nature of C in GaN and they appear when the dopant takes the place of gallium in the crystal structure (*C_Ga_*).

In the case of the GaN:Mn samples, the appearance of mode D was observed. It resembled mixed ${\mathrm{Mn}}^{2+/3+}$ structures observed in Mn_3_O_4_ by Cho et al. [34], who investigated the influence of the Mn oxidation state on the Raman spectrum. In GaN doped with Mn, Mn was present in the neutral ${\mathrm{Mn}}^{3+}$ state with the ${\mathrm{Mn}}^{2+/3+}$ acceptor level pinned at 1.8 eV above the vbm [25]. A strong polarization dependence of band D was similar to the one of the A_1_(LO) mode in GaN [35]. The same band was observed for GaN:Mn,C, for which the position of band D depended on the amount of C (see Table 2 and Figure 2c). The high-frequency shift of this mode with increasing the $\left[C\right]/\left[\mathrm{Mn}\right]$ ratio, presented in Table 3, was very similar to the observation made for GaN:Mn,Mg by Devillers et al. [36] and Nikolenko et al. [37]. In the mentioned studies, the dependence of the position of a mode visible around 667 cm^−1^ was analyzed as a function of the $\left[\mathrm{Mg}\right]/\left[\mathrm{Mn}\right]$ ratio. The observed shifts were explained by changes in the length of the Mn–N bonds, which strongly depended on the occurrence of Mn–Mg complexes. According to T. Devillers et al. [38], the blue shift of mode D can be explained in a way analogous to the shortening of the Mn–N bonds in Mn–C*_k_* (where *k* is an integer equal to 0, 1, 2, or 3) complexes. Using this explanation mode, E can be related to the presence of Mn–C_2_ cation complexes. The origin of the last peak, pinned at 770 cm^−1^, was the ${\mathrm{Mn}}^{4+}$ state [34]. The described spectra showed that for the co-doped samples, C and Mn interacted with each other. It seems that the presence of C in a natural C^4+^ oxidation state forced Mn to also adjust its state to the ${\mathrm{Mn}}^{4+}$. Because of its versatility in the size and a variety of valences, Mn adjusts in size to the presence of C, which is not so flexible.

The LTPL spectra (Figure 3) contained BL and YL for all the measured samples. Some differences between the intensities and positions of these bands can be found. In the C-doped crystals, the BL emission line can be attributed to an electronic transition state from a shallow donor to the (0/+) defect level of isolated $C_{N}$ with energy closer to the vbm than the $C_{N}$ (-/0) level [20]. The most likely source of BL in the examined samples are $C_{N}-H_{i}$ complexes [39]. Another source of this line may be the result of a transition between C incorporated in the gallium site ($C_{Ga}$, which is a shallow donor in this case) and C in the nitrogen site $(C_{N}$). In this case, BL is connected to donor–acceptor pair (DAP) transitions [40,41]. As mentioned, in the case of sample #2, YL is slightly shifted towards higher energies, which is consistent with the results presented in different papers [19,31]. It has already been reported that YL is caused by $C_{N}$ acceptors [42]. It should be noted that for the GaN:Mn samples, YL and BL are clearly seen and they are not caused by C due to its low concentration. According to Reshchikov et al. [40,43] and Xu et al. [44], emission close to 2.4 eV and 2.9 eV can be associated with the presence of isolated gallium vacancies ($V_{Ga}$) or their complexes. The generation of $V_{Ga}$ in the GaN:Mn samples can be correlated to an excessive distortion of the GaN lattice due to the high Mn incorporation [45]. Additionally, found DAP peaks may be caused by transitions of electrons from donors (in the case of HVPE-GaN crystals: Si_Ga_ or O_N_, which are shallow donors) to shallow acceptors [44,46]. In comparison with the GaN:Mn spectra, for GaN:Mn,C the intensity of YL is visibly stronger. This is consistent with the SIMS measurements which show that the C content in the GaN:Mn samples is below the detection limit. The most probable source of YL in GaN:Mn,C are C-related defects, as in the crystals doped only with C. The origin of AL detected in GaN:C with a maximum at 2.59 eV is not clear and requires further investigation [43].

In general, all the LTPL measurements of the investigated SI crystals revealed a strong blue shift of YL in comparison with UID GaN. This behavior is known and was well described by Reshchikov et al. [47]. The shift can be explained by the presence of strong internal electric fields occurring in heavily doped materials. This hypothesis can be supported by the SIMS results which show high doping of all the samples with acceptors. Additionally, a careful analysis of YL allowed to determine its different components. One of them is a line at around 2.36 eV—this band is most often associated with nitrogen vacancies ($V_{N}$) in GaN crystals. Due to low formation energy under p-type conditions for crystals doped with acceptors, it is easy to create $V_{N}$, which are donors in GaN [47].

The analysis of the optical properties of the Mn-doped crystals allowed for an unambiguous determination of the electrical states of Mn in GaN. In the Raman spectrum, the occurrence of the mode at position 667 cm^−1^ is related to ${\mathrm{Mn}}^{2+/3+}$. Manganese builds in as acceptor in place of Ga and forms a simple complex with N. The manganese ${\mathrm{Mn}}^{2+/3+}$ level is located 1.8 eV above vbm [24,26]. In the samples co-doped with C, the Mn electrical state was changed. The difference between the Raman spectra of GaN:Mn and GaN:Mn,C indicates the existence of other states of Mn. The spectra for samples #6, #7, and #8 are very similar to those of the crystals highly doped with Mn and magnesium (Mg) [36,37,38]—except for the appearance of mode F at 770 cm^−1^. Mode E, appearing in the Raman spectra of GaN:Mn,C at around 692 cm^−1^, is related to ${\mathrm{Mn}}^{3+/4+}$. Analyzing the LTPL spectra one can find a line at 3.32 eV, which is assigned to the electron photoionization from the $Mn_{Ga}^{-}$ acceptors. It can be explained by the excitation of ions from ${\mathrm{Mn}}^{2+}$ to the conduction band for photon energy greater than 3.3 eV. A free electron is then trapped at a center associated with ${\mathrm{Mn}}^{4+}$. The trap occupied by this electron captures a hole and recombination excites electrons from the 3d shell of Mn ions, leaving the ${\mathrm{Mn}}^{4+}$ centers in an excited ^4^T_2_(F) state [48]. Thus, for this process to take place, both ${\mathrm{Mn}}^{2+}$ and ${\mathrm{Mn}}^{4+}$ are needed in the structure. The line at 3.32 eV was not observed for GaN:Mn crystals. This means that ${\mathrm{Mn}}^{3+/4+}$ was detected only in the co-doped samples. In HVPE-GaN:Mn, the energy level connected to Mn is placed in the middle of the band gap as ${\mathrm{Mn}}^{2+/3+}$. The ${\mathrm{Mn}}^{3+/4+}$ level lies closer to the vbm than ${\mathrm{Mn}}^{2+/3+}$. No doubt, the different electrical state of Mn detected in the crystals was due to the presence of C. Based on the Raman and LTPL spectra of the samples, it was presented that in the GaN:Mn,C crystals, Mn is in a different electrical state in comparison with Mn in GaN:Mn. Unfortunately, this observation has not been confirmed by electrical measurements. This discrepancy in the results can be explained by the appearance of Mn–C*_k_* complexes, which do not affect the electrical properties of the examined samples—despite the fact that they are visible in the optical measurements.

As the Hall measurements showed, the co-doped crystals exhibit p-type conductivity. This is in contrary to GaN:Mn, which is n-type at high temperature [15]. As mentioned, C in GaN can be built as C_N_ (acceptor) or *C_Ga_* (donor). The amphoteric nature of C is well known and was thoroughly studied [20,21,22,23,33,49]. The above-discussed optical measurements also confirm this behavior.

The results obtained from the fitting of the electrical data lead to very interesting conclusions. Firstly, all the data can be described by the same Mn ionization energy (*E_Mn_*) of 1.5 eV above vbm. It corresponds to the known ${\mathrm{Mn}}^{2+/3+}$ transfer level [25]. In the case of p-type samples (highly doped GaN:Mn and GaN:Mn,C), this level is responsible for the lower activation energy describing the slopes of the p(T) and ρ(T) curves than in the case of n-type GaN:Mn crystals (1.8 eV) [16]. Simply, for such a position in the bandgap, the transfer of electrons from the valence band to this level requires lower energy than the generation of electrons to the conduction band. Contrary to the n-type crystals, in the p-type samples, the Fermi level is slightly below the ${\mathrm{Mn}}^{2+/3+}$ one, and all the Mn ions are in the Mn^3+^ state at low temperature (the ${\mathrm{Mn}}^{2+/3+}$ level is unoccupied). Secondly, it seems that the electrical properties of GaN:Mn are almost unaffected by the presence of C. It can be seen that for a lower C content, the resistivity of the material increases (sample #6) in comparison with ref_Mn [15]. On the other hand, for highly doped samples, the resistivity decreases and reaches the same values as sample #8. In order to describe the data, it has to be assumed that the total concentration of donors is equal to the total concentration of acceptors (excluding Mn). Since the residual donor content does not exceed 2 × 10^17^ cm^−3^, a predominant donor contribution comes from the defects involving *C_Ga_*. On the other hand, if all C detected by SIMS acted as C_N_ acceptors, the resulting hole concentration would be much larger than revealed by the electrical measurements, as shown by the blue line in Figure 4a. Thus, in order to fit the data in the best way, the ratio of donors to C acceptors (excluding Mn) of 1:1 had to be assumed. In other words, the same results would be obtained if we neglected the presence of C and donors at all. This result is consistent with data published in [23], where for a high C concentration (10^19^ cm^−3^), the compensation ratio reached 1. At a carbon concentration of 1 × 10^18^ cm^−3^, this ratio was around 60%.

Carbon seems to be, somehow, responsible for the p-type conductivity of HVPE-GaN:Mn,C at high temperatures. HVPE-GaN doped only with Mn predominantly showed n-type conductivity at high temperatures. However, if all C atoms determined by SIMS acted as acceptors, the hole concentration would be much higher than that measured (see Figure 4a and the blue line). The C acceptors are, therefore, entirely compensated by C donors. The observed hole activation is connected to the ${\mathrm{Mn}}^{2+/3+}$ level. This case is similar to high nitrogen pressure solution (HNPS) GaN samples doped with Mg (HNPS-GaN:Mg), where Mg was totally compensated with oxygen atoms. The energy level connected to *V_Ga_* was observed at around 1 eV above vbm, between the Mg and oxygen levels [50]. One can risk a statement that a high concentration of C does not introduce any significant changes to the electrical properties of GaN because C compensates itself. The compensation ratio close to 1 gives the best fit for resistivity as a function of inverse temperature.

## 5. Conclusions

This work presents the feasibility to intentionally crystallize HVPE-GaN co-doped with Mn and C. It is possible to control the concentrations of the dopants in the crystals by changing the flows of dopant precursors in the HVPE system. The structural, as well as optical and electrical, properties of the obtained crystals were examined. Based on the optical properties of the samples, it was presented that in the GaN:Mn,C crystals, Mn is in a different electrical state in comparison with Mn in GaN:Mn. This was, however, not confirmed by the electrical measurements of the obtained samples. Investigation of electrical properties showed that C can change the type of conductivity in the new-grown co-doped crystals. All the conducted research shows that HVPE-GaN:Mn or co-doped HVPE-GaN:Mn,C are good candidates for fabricating SI GaN wafers. In the next step, mechanical properties of GaN:Mn,C and their influence on wafering the crystals should be examined.

## Figures and Tables

**Figure 1 materials-15-02379-f001:**
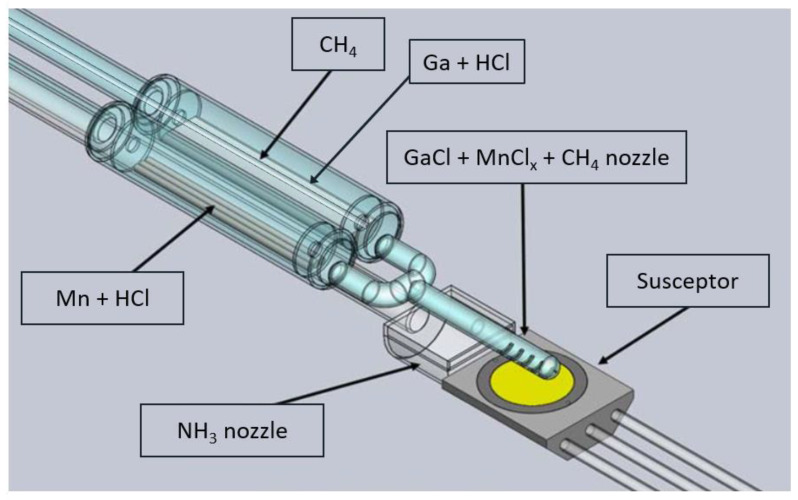
Scheme of the HVPE reactor configuration; GaCl, CH_4_, and MnCl_x_ tubes were connected into one, allowing the reactants to mix and reach the growth zone by one quartz nozzle.

**Figure 2 materials-15-02379-f002:**
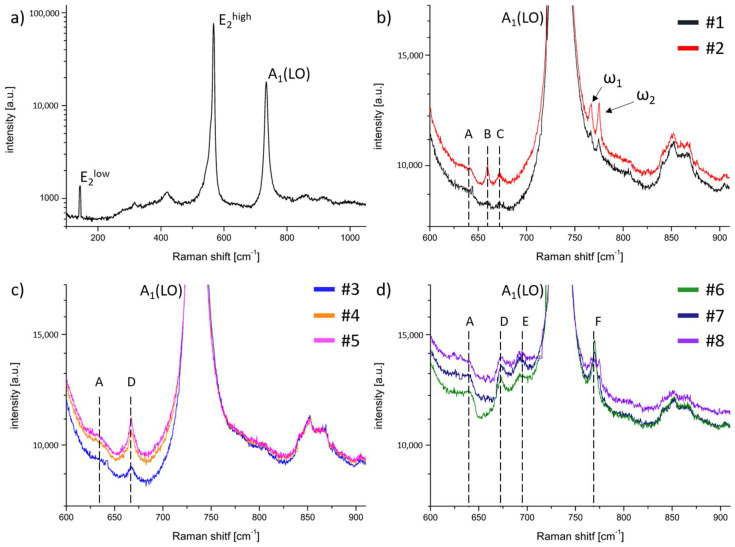
Raman spectra for (**a**) UID-HVPE-GaN; (**b**) HVPE-GaN:C; (**c**) HVPE-GaN:Mn; and (**d**) HVPE-GaN:Mn,C collected at ambient conditions in $z\left(xx\right)\bar{z}$ configuration.

**Figure 3 materials-15-02379-f003:**
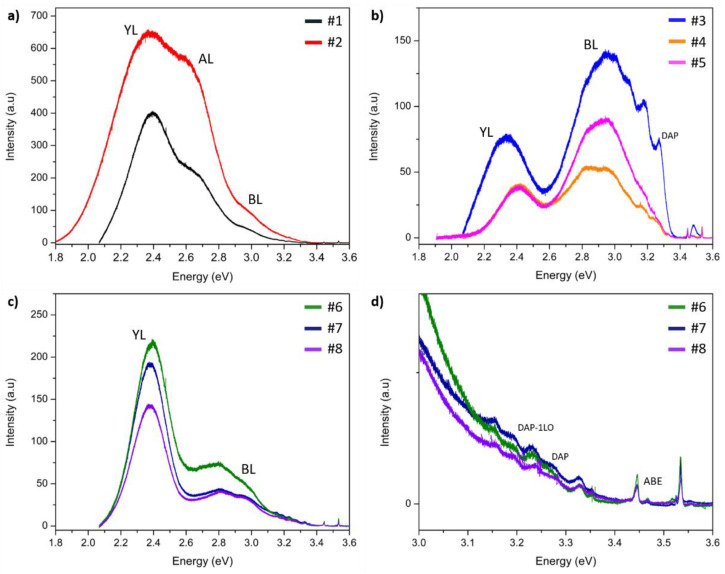
LTPL spectra for HVPE samples: (**a**) GaN:C; (**b**) GaN:Mn; (**c**) GaN:Mn,C; and (**d**) GaN:Mn,C in the range of 3.0 eV–3.6 eV.

**Table 1 materials-15-02379-t001:** Technological parameters applied for growth of HVPE-GaN:C, HVPE-GaN:Mn, and HVPE-GaN:Mn,C.

	GaN:C		GaN:Mn			GaN:Mn,C		
	#1	#2	#3	#4	#5	#6	#7	#8
CH_4_ (ml/min)	10	20	0	0	0	3	10	20
HCl over Mn (ml/min)	0	0	0.2	0.3	0.6	0.3	0.3	0.3
HCl over Ga (ml/min)	48	48	48	48	48	48	48	48

**Table 2 materials-15-02379-t002:** SIMS data for $\left(0001\right)$ surfaces of the samples.

No.	(C) cm^−3^	(Mn) cm^−3^	(O) cm^−3^	(Si) cm^−3^
#1	1.5 × 10^19^	<5 × 10^15^	1 × 10^17^	1 × 10^17^
#2	4 × 10^19^	<5 × 10^15^	7 × 10^16^	1 × 10^17^
#3	<2 × 10^16^	2.5 × 10^18^	9 × 10^16^	1 × 10^17^
#4	<2 × 10^16^	3 × 10^18^	8 × 10^16^	1 × 10^17^
#5	<2 × 10^16^	5 × 10^18^	1 × 10^17^	1 × 10^17^
#6	5 × 10^18^	3 × 10^18^	1 × 10^17^	8 × 10^16^
#7	2 × 10^19^	2.5 × 10^18^	1 × 10^17^	1 × 10^17^
#8	4.5 × 10^19^	3 × 10^18^	9 × 10^16^	1.5 × 10^17^

**Table 3 materials-15-02379-t003:** Ratio $\left[C\right]/\left[\mathrm{Mn}\right]\;$ and position of D peak for all samples.

	#1	#2	#3	#4	#5	#6	#7	#8
$\left[C\right]/\left[\mathrm{Mn}\right]\;$	-	-	0	0	0	1.67	8	15
ωD [cm−1]	-	-	667	667	667	672.5	673.5	674.5

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
