# Peer review of "Carbon and Manganese in Semi-Insulating Bulk GaN Crystals"

_materials, 2022, doi:10.3390/ma15072379_

Round 1
Reviewer 1 Report
Article “Carbon and Manganese in Semi-insulating Bulk GaN Crystals” is devoted to GaN co-doping with manganese and carbon preparation. The article presents interesting scientific resultsÑŽ
To improve the article, I recommend that the authors pay attention to the following comments:
- In the introduction, the purpose of the work should be more clearly formulated.
- Line 87 says "A quartz, horizontal home-made HVPE reactor...". Quartz is a crystalline mineral of silicon(IV) oxide. Most likely, the authors mean silica glass reactor.
- How did the authors confirm the formation of GaCl by passing hydrogen chloride over gallium? The thermodynamically most probable product of the interaction of gallium with hydrogen chloride at low temperatures is gallium(III) chloride. GaCl can be formed at high temperatures (1370 K) in the GaN synthesis reactor; however, the flow of gallium into the reactor, judging by the scheme in Fig. 1, still occurs in the form of GaCl3. This requires clarification.
- I think that in Table 2, instead of BGL, it is better to indicate the values ​​<5×1015 and <2×1016, because for manganese and carbon, the BGLs are different.
I think that after revision the article can be published.

Author Response
Dear Reviewer,
Below, please find the list of corrections according to your remarks.
In the introduction, the purpose of the work should be more clearly formulated.
The Reviewer is right, the goal behind this work was not clearly formulated. We propose to slightly change the following part (lines 69-74):
“However, co-doping with C could change some mechanical properties of SI HVPE-GaN and facilitate the wafering procedures. Before checking the mechanical properties and launching the fabrication of new coË—doped GaN wafers, physical (structural, optical, and electrical) properties of HVPEË—GaN:Mn,C should be examined and analyzed. Therefore, a co-doping process of HVPEË—GaN by Mn and C was performed.”
to:
“On the other hand, according to theoretical predictions [26], Mn may also exist in the Mn3+/4+ level. This state can, theoretically, be obtained by co-doping of Mn with other acceptors. However, the existence of Mn3+/4+ in bulk GaN crystals has not been confirmed experimentally so far. In this work Mn3+/4+ will be detected in GaN doped with Mn and C. Therefore, a co-doping process of HVPEË—GaN by Mn and C was performed. Moreover, co-doping with C could change some mechanical properties of SI HVPE-GaN and facilitate the wafering procedures. Before checking the mechanical properties and launching the fabrication of new coË—doped GaN wafers, physical (structural, optical, and electrical) properties of HVPEË—GaN:Mn,C should be examined and analyzed.”
Line 87 says "A quartz, horizontal home-made HVPE reactor...". Quartz is a crystalline mineral of silicon(IV) oxide. Most likely, the authors mean silica glass reactor.
The Reviewer is right. The word "quartz" has been replaced with "silica glass".
How did the authors confirm the formation of GaCl by passing hydrogen chloride over gallium? The thermodynamically most probable product of the interaction of gallium with hydrogen chloride at low temperatures is gallium(III) chloride. GaCl can be formed at high temperatures (1370 K) in the GaN synthesis reactor; however, the flow of gallium into the reactor, judging by the scheme in Fig. 1, still occurs in the form of GaCl3. This requires clarification.
The Reviewer is right that both GaCl and GaCl3 can be formed as a result of the reaction of Ga with HCl. However, according to the theoretical work conducted by A. Koukitu and Y. Kumagai [Koukitu, A. and Kumagai, Y. (2010). Technology of Gallium Nitride Crystal Growth (eds. D. Ehrentraut, E. Meissner and M. Bockowski), 31. Heidelberg: Springer-Verlag] as well as by P. Kempisty [Journal of Crystal Growth 296 (2006) 31-42] in the conditions we use, the partial pressure of GaCl is much higher than that of GaCl3. Therefore, formation of GaCl is more favorable. We predict that during the experiment, mainly GaCl is synthesized.
We propose to add more information about HVPE method:
“The HVPE method is the crystallization from gas phase. Hydrochloride reacts with liquid gallium at relatively low temperature (800–900 °C) forming gallium chloride (GaCl). In this temperature range, the partial pressure of GaCl is much higher than that of GaCl3. Therefore, formation of GaCl is more favorable [27]. The latter is transported by the carrier gas (CG, mainly hydrogen or nitrogen) to the crystal growth zone (at temperature of 1000–1100 °C) where a seed is placed. Herein, GaCl reacts with ammonia (NH3) to synthesize GaN. A silica glass, horizontal home…”
I think that in Table 2, instead of BGL, it is better to indicate the values ​​<5×1015 and <2×1016, because for manganese and carbon, the BGLs are different.
We thank the Reviewer for this valuable remark. “BGL” was changed to proposed values (​​<5×1015 and <2×1016).
We would like to thank for your corrections which really helped us improve the manuscript.

Reviewer 2 Report
The authors provide very useful information on the preparation oflayers suitable as substrates for GaN growth.
The optical characterization of the prepared layers is very interesting
and brings new information, but the goal behind which they
do this is not clearly proven.
Please, give more information about contact quality for Hall measurement.
Please give more information about the profile of doping, it is very important
in the case of slicing of samples. In my opinion it would be very useful
to bring SIMS profile, or bring the optical measurement results from the
back side of sample prepared.
Author Response
Dear Reviewer,
Below, please find the list of corrections according to your remarks.
The authors provide very useful information on the preparation of layers suitable as substrates for GaN growth. The optical characterization of the prepared layers is very interesting and brings new information, but the goal behind which they do this is not clearly proven.
The Reviewer is right, the purpose of this work was not clearly formulated and it seems to be the goal is not clearly proven. Therefore, we propose slightly change the following part (lines 69-74):
“However, co-doping with C could change some mechanical properties of SI HVPE-GaN and facilitate the wafering procedures. Before checking the mechanical properties and launching the fabrication of new coË—doped GaN wafers, physical (structural, optical, and electrical) properties of HVPEË—GaN:Mn,C should be examined and analyzed. Therefore, a co-doping process of HVPEË—GaN by Mn and C was performed.”
to:
“On the other hand, according to theoretical predictions [26], Mn may also exist in the Mn3+/4+ level. This state can, theoretically, be obtained by co-doping of Mn with other acceptors. However, the existence of Mn3+/4+ in bulk GaN crystals has not been confirmed experimentally so far. In this work Mn3+/4+ will be detected in GaN doped with Mn and C. Therefore, a co-doping process of HVPEË—GaN by Mn and C was performed. Moreover, co-doping with C could change some mechanical properties of SI HVPE-GaN and facilitate the wafering procedures. Before checking the mechanical properties and launching the fabrication of new coË—doped GaN wafers, physical (structural, optical, and electrical) properties of HVPEË—GaN:Mn,C should be examined and analyzed.”
Please, give more information about contact quality for Hall measurement.
The Reviewer is right. We propose to add more information about Hall measurements and contacts to the manuscript. After change, this part of manuscript looks as follow:
“Resistivity ρ and Hall constant were measured in the van der Pauw configuration at temperature reaching 1000 K. The samples were prepared by placing Ni(250 Å)/Au(750Å) contacts on the (0001) surface and annealed at 773K for a few minutes under atmosphere of N2 with a 20% admixture of O2. The quality of the contacts enabled measurements of ohmic resistivity in the studied temperature range. The measurement parameters, like operating current, were controlled to ensure ohmic conditions.”
Please give more information about the profile of doping, it is very important in the case of slicing of samples. In my opinion it would be very useful to bring SIMS profile, or bring the optical measurement results from the back side of sample prepared.
We would like to thank the Reviewer for this remark. For the Reviewer’s eyes we present in the attached file table the results of SIMS measurements on the (0001) and (000-1) planes. Typical in-depth profiles measured on these two planes are also presented in the attached file.
We would like to thank for your corrections which really helped us improve the manuscript.

Reviewer 3 Report
see attached file

Author Response
Dear Reviewer,
Below, please find the list of corrections according to your remarks.
-lines 36, 37 (2), 74, 293, 350, 438: SI ?
We thank the Reviewer for this remark. SI means semi-insulating. This acronym is introduced in line 25: “… semi-insulating (SI) substrate as silicon carbide (SiC) or …”
- line 78 : Acronym UID, BGL (table 2)
We thank the Reviewer for this remark. Acronym UID (unintentionally doped) is introduced in lines 50-51: “Mechanical properties of HVPE-GaN:C are similar to those of unintentionally doped (UID) HVPE-GaN…”
Acronym BGL (beckground level) was introduced in line 162 (below Tab.2): “…no traces of Mn were detected (the value was below the background level – BGL…” However, due to the different background levels of carbon and manganese to which the acronym was referred to, we decided to remove this acronym. Table 2 was corrected by introducing values of the background level instead of “BGL”.
- line 258,5 ; equation (2) : please define and quote gi in equation
The Reviewer is right, gi was not defined. Gi in equation (2) is the degeneracy of energy level i. We propose to add this information in the text below equation (2):
“…where i stands for Mn or CN, gi is the degeneracy of energy level i and EF is the Fermi energy, kB –is the Boltzmann constant, and T is temperature.”
- line 309 : vbm acronym
We thank the Reviewer for this remark. Here “vbm” is the valence band maximum. This acronym is introduced in the Introduction section (line 54):
“…at around 1 eV above the valence band maximum (vbm) …”
- What is the limit of detection of H by SIMS, please as H2 is the carrier gas.
Background level of H by SIMS is 1x1016 cm-3.
We propose to add information about background level of hydrogen, silicon and oxygen (lines 173-176):
“The concentrations of the main donors ([Si], oxygen [O] as well as hydrogen [H]) were 1×1017 cmË—3 or lower (the background level of [Si], [O], and [H] is 5×1016 cm-3, 1×1016 cmË—3, and 1×1016 cm-3, respectively).”
We thank the Reviewer for kind words regarding our manuscript.
We would like to thank for your corrections which really helped us improve the manuscript.
